# All Tokens Matter: Token Labeling for Training Better Vision Transformers

**Zihang Jiang**[1][*]   **Qibin Hou**[2,1][†]   **Li Yuan**[3]   **Daquan Zhou**[1]   **Yujun Shi**[1]

**Xiaojie Jin**[4]   **Anran Wang**[4]   **Jiashi Feng**[4]

[1]**National University of Singapore**   [2]**Nankai University**

[3] **Peking University**   [4]**ByteDance**

{jzh0103,andrewhoux,ylustcnus,zhoudaquan21,shiyujun1016}@gmail.com
xjjin0731@gmail.com, anran.wang@bytedance.com, jshfeng@gmail.com

## Abstract

In this paper, we present token labeling—a new training objective for training high-performance vision transformers (ViTs). Different from the standard training objective of ViTs that computes the classification loss on an additional trainable class token, our proposed one takes advantage of all the image patch tokens to compute the training loss in a dense manner. Specifically, token labeling reformulates the image classification problem into multiple token-level recognition problems and assigns each patch token with an individual location-specific supervision generated by a machine annotator. Experiments show that token labeling can clearly and consistently improve the performance of various ViT models across a wide spectrum. For a vision transformer with 26M learnable parameters serving as an example, with token labeling, the model can achieve 84.4% Top-1 accuracy on ImageNet. The result can be further increased to 86.4% by slightly scaling the model size up to 150M, delivering the minimal-sized model among previous models (250M+) reaching 86%. We also show that token labeling can clearly improve the generalization capability of the pretrained models on downstream tasks with dense prediction, such as semantic segmentation. Our code and model are publicly available at https://github.com/zihangJiang/TokenLabeling.

## 1 Introduction

Transformers [39] have achieved great performance for almost all the natural language processing (NLP) tasks over the past years [4, 14, 24]. Motivated by such success, recently, many researchers attempt to build transformer models for vision tasks, and their encouraging results have shown the great potential of transformer based models for image classification [6, 15, 25, 36, 40, 46], especially the strong benefits of the self-attention mechanism in building long-range dependencies between pairs of input tokens.

Despite the importance of gathering long-range dependencies, recent work on local data augmentation [57] has demonstrated that well modeling and leveraging local information for image classification would avoid biasing the model towards skewed and non-generalizable patterns and substantially

---

[*]Work done as an intern at ByteDance AI Lab.
[†]Corresponding author. Part of this work was done as a research fellow at NUS.

35th Conference on Neural Information Processing Systems (NeurIPS 2021).

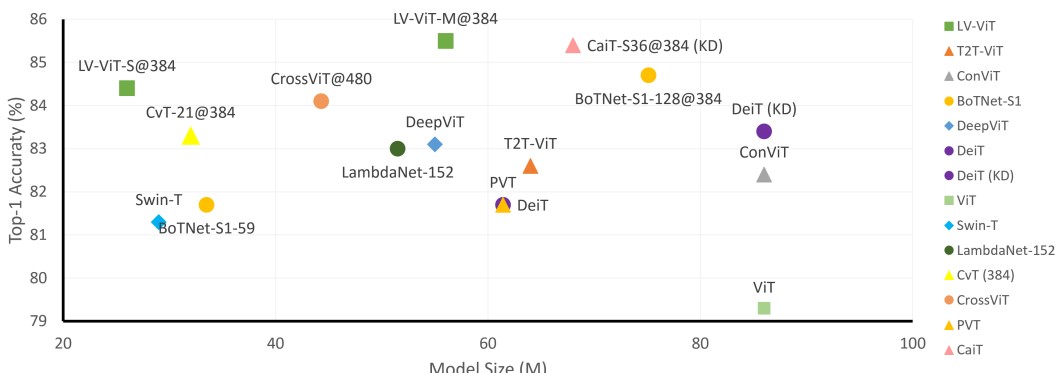

Figure 1: Comparison between the proposed LV-ViT and other recent works based on vision transformers, including T2T-ViT [46], ConViT [12], BoTNet [31], DeepViT [59], DeiT [36], ViT [15], Swin Transformer [25], LambdaNet [1], CvT [43], CrossViT [6], PVT [40], CaiT [37]. Note that we only show models whose model sizes are under 100M. As can be seen, our LV-ViT achieves the best results using the least amount of learnable parameters. The default test resolution is $224 \times 224$ unless specified after @.

improve the model performance. However, recent vision transformers normally utilize class tokens that aggregate global information to predict the output class while neglecting the role of other patch tokens that encode rich information on their respective local image patches.

In this paper, we present a new training objective for vision transformers, termed *token labeling*, that takes advantage of both the patch tokens and the class tokens. Our method takes a $K$-dimensional score map generated by a machine annotator as supervision to supervise all the tokens in a dense manner, where $K$ is the number of categories for the target dataset. In this way, each patch token is explicitly associated with an individual location-specific supervision indicating the existence of the target objects inside the corresponding image patch, so as to improve the object grounding and recognition capabilities of vision transformers with negligible computation overhead. To the best of our knowledge, this is the first work demonstrating that dense supervision is beneficial to vision transformers in image classification.

According to our experiments, utilizing the proposed token labeling objective can clearly boost the performance of vision transformers. As shown in Figure 1, our model, named LV-ViT, with 56M parameters, yields 85.4% top-1 accuracy on ImageNet [13], behaving better than all the other transformer-based models having no more than 100M parameters. When the model size is scaled up to 150M, the result can be further improved to 86.4%. In addition, we have empirically found that the pretrained models with token labeling are also beneficial to downstream tasks with dense prediction, such as semantic segmentation.

## 2 Related Work

Transformers [39] refer to the models that entirely rely on the self-attention mechanism to build global dependencies, which are originally designed for natural language processing tasks. Due to their strong capability of capturing spatial information, transformers have also been successfully applied to a variety of vision problems, including low-level vision tasks like image enhancement [7, 45], as well as more challenging tasks such as image classification [9, 15], object detection [5, 11, 55, 61], segmentation [7, 33, 41] and image generation [28]. Some works also extend transformers for video and 3D point cloud processing [50, 53, 60].

Vision Transformer (ViT) is one of the earlier attempts that achieved state-of-the-art performance on ImageNet classification, using pure transformers as basic building blocks. However, ViTs need pretraining on very large datasets, such as ImageNet-22k and JFT-300M, and huge computation resources to achieve comparable performance to ResNet [18] with a similar model size trained on ImageNet. Later, DeiT [36] manages to tackle the data-inefficiency problem by simply adjusting the network architecture and adding an additional token along with the class token for Knowledge Distillation [21, 47] to improve model performance.

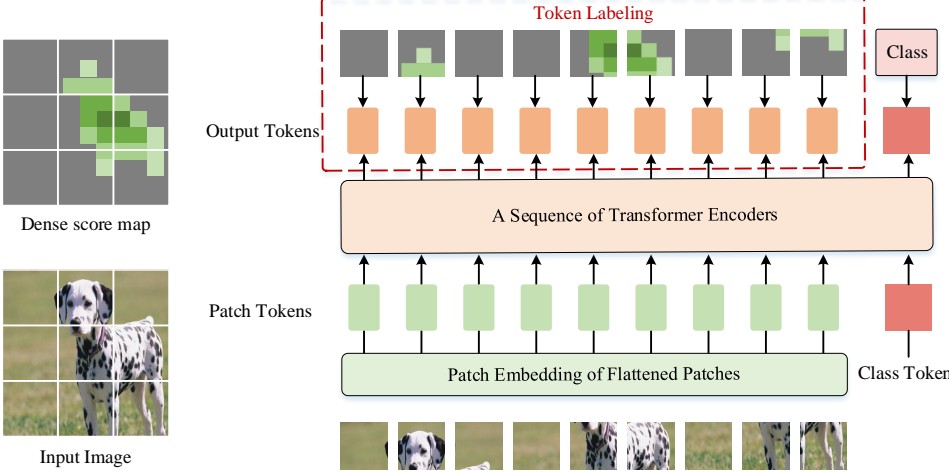

Figure 2: Pipeline of training vision transformers with token labeling. Other than utilizing the class token (pink rectangle), we also take advantage of all the output patch tokens (orange rounded rectangle) by assigning each patch token an individual location-specific prediction generated by a machine annotator [3] as supervision (see the part in the red dash rectangle). Our proposed token labeling method can be treated as an auxiliary objective to provide each patch token the local details that aid vision transformers to more accurately locate and recognize the target objects. Note that the traditional vision transformer training does not include the red dash rectangle part.

Some recent works [6, 16, 43, 46] also attempt to introduce the local dependency into vision transformers by modifying the patch embedding block or the transformer block or both, leading to significant performance gains. Moreover, there are also some works [20, 25, 40] adopting a pyramid structure to reduce the overall computation while maintaining the model's ability to capture low-level features.

Unlike most aforementioned works that design new transformer blocks or transformer architectures, we attempt to improve vision transformers by studying the role of patch tokens that embed rich local information inside image patches. We show that by slightly tuning the structure of vision transformers and employing the proposed token labeling objective, we can achieve strong baselines for transformer models at different model size levels.

## 3 Token Labeling Method

In this section, we first briefly review the structure of the vision transformer [15] and then describe the proposed training objective—*token labeling*.

### 3.1 Revisiting Vision Transformer

A typical vision transformer [15] first decomposes a fixed-size input image into a sequence of small patches. Each small patch is mapped to a feature vector, or called a token, by projection with a linear layer. Then, all the tokens combined with an additional learnable class token for classification score prediction are sent into a stack of transformer blocks for feature encoding.

In loss computing, the class token from the output tokens of the last transformer block is usually selected and sent into a linear layer for the classification score prediction. Mathematically, given an image $I$, denote the output of the last transformer block as $[X^{cls}, X^1, ..., X^N]$, where $N$ is the total number of patch tokens, and $X^{cls}$ and $X^1, ..., X^N$ correspond to the class token and the patch tokens, respectively. The classification loss for image $I$ can be written as

$$L_{cls} = H(X^{cls}, y^{cls}), \tag{1}$$

where $H(\cdot, \cdot)$ is the softmax cross-entropy loss and $y^{cls}$ is the class label.

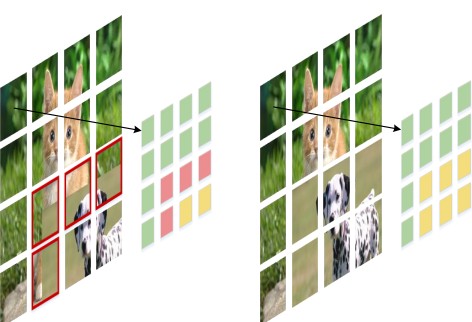

Figure 3: Comparison between CutMix [48] (**Left**) and our proposed MixToken (**Right**). CutMix is operated on the input images. This results in patches containing mixed regions from the two images (see the patches enclosed by red bounding boxes). Differently, MixToken targets at mixing tokens after patch embedding. This enables each token after patch embedding to have clean content as shown in the right part of this figure. The detailed advantage of MixToken can be found in Sec. 4.2.

### 3.2 Token Labeling

The above classification problem only adopts an image-level label as supervision whereas it neglects the rich information embedded in each image patch. In this subsection, we present a new training objective—*token labeling*—that takes advantage of the complementary information between the patch tokens and the class tokens.

**Token Labeling:** Different from the classification loss as formulated in Eqn. (1) that measures the distance between the single class token (representing the whole input image) and the corresponding image-level label, token labeling emphasizes the importance of all output tokens and advocates that each output token should be associated with an individual location-specific label. Therefore, in our method, the ground truth for an input image involves not only a single $K$-dimensional vector $y^{cls}$ but also a $K \times N$ matrix or called a $K$-dimensional score map as represented by $[y^1, ..., y^N]$, where $N$ is the number of the output patch tokens.

Specifically, we leverage a dense score map for each training image and use the cross-entropy loss between each output patch token and the corresponding aligned label in the dense score map as an auxiliary loss at the training phase. Figure 2 provides an intuitive interpretation. Given the output patch tokens $X^1, ..., X^N$ and the corresponding labels $[y^1, ..., y^N]$, the token labeling objective can be defined as

$$L_{tl} = \frac{1}{N} \sum_{i=1}^{N} H(X^i, y^i). \tag{2}$$

Recall that $H$ is the cross-entropy loss. Therefore, the total loss function can be written as

$$L_{total} = H(X^{cls}, y^{cls}) + \beta \cdot L_{tl}, \tag{3}$$

$$= H(X^{cls}, y^{cls}) + \beta \cdot \frac{1}{N} \sum_{i=1}^{N} H(X^i, y^i), \tag{4}$$

where $\beta$ is a hyper-parameter to balance the two terms. In our experiment, we empirically set it to 0.5.

**Advantages:** Our token labeling offers the following advantages. *First of all*, unlike knowledge distillation methods that require a teacher model to generate supervision labels online, token labeling is a cheap operation. The dense score map can be generated by a pretrained model in advance (e.g., EfficientNet [34] or NFNet [3]). During training, we only need to crop the score map and perform interpolation to make it aligned with the cropped image in the spatial coordinate. Thus, the additional computations are negligible. *Second*, rather than utilizing a single label vector as supervision as done in most classification models and the ReLabel strategy [49], we also harness score maps to supervise the models in a dense manner and thereby the label for each patch token provides location-specific information, which can aid the training models to easily discover the target objects and improve the recognition accuracy. *Last but not the least*, as dense supervision is adopted in training, we found that the pretrained models with token labeling benefit downstream tasks with dense prediction, like semantic segmentation.

### 3.3 Token Labeling with MixToken

While training vision transformer, previous studies [36, 46] have shown that augmentation methods, like MixUp [52] and CutMix [48], can effectively boost the performance and robustness of the models. However, vision transformers rely on patch-based tokenization to map each input image to a sequence of tokens and our token labeling strategy also operates on patch-based token labels. If we apply CutMix directly on the raw image, some of the resulting patches may contain content from two images, leading to mixed regions within a small patch as shown in Figure 3. When performing token labeling, it is difficult to assign each output token a clean and correct label. Taking this situation into account, we rethink the CutMix augmentation method and present MixToken, which can be viewed as a modified version of CutMix operating on the tokens after patch embedding as illustrated in the right part of Figure 3.

To be specific, for two images denoted as $I_1$, $I_2$ and their corresponding token labels $Y_1 = [y_1^1, ..., y_1^N]$ as well as $Y_2 = [y_2^1, ..., y_2^N]$, we first feed the two images into the patch embedding module to tokenize each as a sequence of tokens, resulting in $T_1 = [t_1^1, ..., t_1^N]$ and $T_2 = [t_2^1, ..., t_2^N]$. Then, we produce a new sequence of tokens by applying MixToken using a binary mask $M$ as follows:

$$\hat{T} = T_1 \odot M + T_2 \odot (1 - M), \tag{5}$$

where $\odot$ is element-wise multiplication. We use the same way to generate the mask $M$ as in [48]. For the corresponding token labels, we also mix them using the same mask $M$:

$$\hat{Y} = Y_1 \odot M + Y_2 \odot (1 - M). \tag{6}$$

The label for the class token can be written as

$$\hat{y^{cls}} = \bar{M} y_1^{cls} + (1 - \bar{M}) y_2^{cls}, \tag{7}$$

where $\bar{M}$ is the average of all element values of $M$.

## 4 Experiments

### 4.1 Experiment Setup

We evaluate our method on the ImageNet [13] dataset. All experiments are built and conducted upon `PyTorch` [29] and the `timm` [42] library. We follow the standard training schedule and train our models on the ImageNet dataset for 300 epochs. Besides normal augmentations like CutOut [57] and RandAug [10], we also explore the effect of applying MixUp [52] and CutMix [48] together with our proposed token labeling. Empirically, we have found that using MixUp together with token labeling brings no benefit to the performance, and thus we do not apply it in our experiments.

For optimization, by default, we use the AdamW optimizer [27] with a linear learning rate scaling strategy $lr = 10^{-3} \times \frac{batch\_size}{640}$ and $5 \times 10^{-2}$ weight decay rate. For Dropout regularization, we observe that for small models, using Dropout hurts the performance. This has also been observed in a few other works related to training vision transformers [36, 37, 46]. As a result, we do not apply Dropout [32] and use Stochastic Depth [23] instead. More details on hyper-parameters and finetuning can be found in our supplementary materials.

We use the NFNet-F6 [3] trained on ImageNet with an $86.3\%$ Top-1 accuracy as the machine annotator to generate dense score maps for the ImageNet dataset, yielding a 1000-dimensional score map for each image for training. The score map generation procedure is similar to [49], but we limit our experiment setting by training all models from scratch on ImageNet without extra data support, such as JFT-300M and ImageNet-22K. This is different from the original ReLabel paper [49], in which the EfficientNet-L2 model pretrained on JFT-300M is used. The input resolution for NFNet-F6 is $576 \times 576$, and the dimension of the corresponding output score map for each image is $L \in \mathbb{R}^{18 \times 18 \times 1000}$. During training, the target labels for the tokens are generated by applying RoIAlign [17] on the corresponding score map. In practice, we only store the top-5 score maps for each position in half-precision to save space as storing the entire score maps for all the images results in 2TB storage. In our experiment, we only need 10GB of storage to store all the score maps.

Table 1: Performance of the proposed LV-ViT with different model sizes. Here, 'depth' denotes the number of transformer blocks used in different models. By default, the test resolution is set to $224 \times 224$ except the last one which is $288 \times 288$.

| Name | Depth | Embed dim. | MLP Ratio | #Heads | #Params | Throughput (im/s) | Test size | Top-1 Acc. (%) |
|------|-------|-----------|-----------|--------|---------|-------------------|-----------|----------------|
| LV-ViT-T | 12 | 240 | 3.0 | 4 | 8.5M | 2032.6 | 224 | 79.1 |
| LV-ViT-S | 16 | 384 | 3.0 | 6 | 26M | 1018.2 | 224 | 83.3 |
| LV-ViT-M | 20 | 512 | 3.0 | 8 | 56M | 668.9 | 224 | 84.1 |
| LV-ViT-L | 24 | 768 | 3.0 | 12 | 150M | 204.8 | 288 | **85.3** |

## 4.2 Ablation Analysis

**Model Settings:** The default settings of the proposed LV-ViT are given in Table 1, where both token labeling and MixToken are used. A slight architecture modification to ViT [15] is that we replace the patch embedding module with a 4-layer convolution to better tokenize the input image and integrate local information. Detailed ablation about patch embedding can be found in our supplementary materials. As can be seen, our LV-ViT-T with only 8.5M parameters can already achieve a top-1 accuracy of 79.1% on ImageNet. Increasing the embedding dimension and network depth can further boost the performance. More experiments compared to other methods can be found in Sec. 4.3. In the following ablation experiments, we will set our LV-ViT-S as baseline and show the advantages of the proposed token labeling and MixToken methods.

**MixToken:** We use MixToken as a substitution for CutMix while applying token labeling. Our experiments show that MixToken performs better than CutMix for token-based transformer models. As shown in Table 2, when training with the original ImageNet labels, using MixToken is 0.1% higher than using CutMix. When using the ReLabel supervision, we can also see an advantage of 0.2% over the CutMix baseline. Combining with our token labeling, the performance can be further raised to 83.3%.

Table 2: Ablation on the proposed MixToken and token labeling augmentations. We also show results with either the ImageNet hard label and the ReLabel [49] as supervision.

| Aug. Method | Supervision | Top-1 Acc. |
|-------------|-------------|------------|
| MixToken | Token labeling | **83.3** |
| MixToken | ReLabel | 83.0 |
| CutMix | ReLabel | 82.8 |
| Mixtoken | ImageNet Label | 82.5 |
| CutMix | ImageNet Label | 82.4 |

Table 3: Ablation on different widely-used data augmentations. We have empirically found our proposed MixToken performs even better than the combination of MixUp and CutMix in vision transformers.

| MixToken | MixUp | CutOut | RandAug | Top-1 Acc. |
|----------|-------|--------|---------|------------|
| ✓ | ✗ | ✓ | ✓ | **83.3** |
| ✗ | ✗ | ✓ | ✓ | 81.3 |
| ✓ | ✓ | ✓ | ✓ | 83.1 |
| ✓ | ✗ | ✗ | ✓ | 83.0 |
| ✓ | ✗ | ✓ | ✗ | 82.8 |

**Data Augmentation:** Here, we study the compatibility of MixToken with other augmentation techniques, such as MixUp [52], CutOut [57] and RandAug [10]. The ablation results are shown in Table 3. We can see when all the four augmentation methods are used, a top-1 accuracy of 83.1% is achieved. Interestingly, when the MixUp augmentation is removed, the performance can be improved to 83.3%. This may be explained as, using MixToken and MixUp at the same time would bring too much noise in the label, and consequently cause confusion of the model. Moreover, the CutOut augmentation, which randomly erases some parts of the image, is also effective and removing it brings a performance drop of 0.3%. Similarly, the RandAug augmentation also contributes to the performance and using it brings an improvement of 0.5%.

**All Tokens Matter:** To show the importance of involving all tokens in our token labeling method, we attempt to randomly drop some tokens and use the remaining ones for computing the token labeling loss. The percentage of the remaining tokens is denoted as Token Participation Rate. As shown in Figure 4 (Left), we conduct experiments on two models: LV-ViT-S and LV-ViT-M. As can be seen, using only 20% of the tokens to compute the token labeling loss decreases the performance ($-0.5\%$ for LV-ViT-S and $-0.4\%$ for LV-ViT-M). Involving more tokens for loss computation consistently leads to better performance. Since involving all tokens brings negligible computation cost and gives the best performance, we always set the token participation rate as 100% in the following experiments.

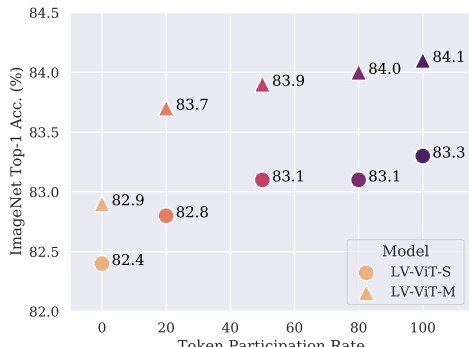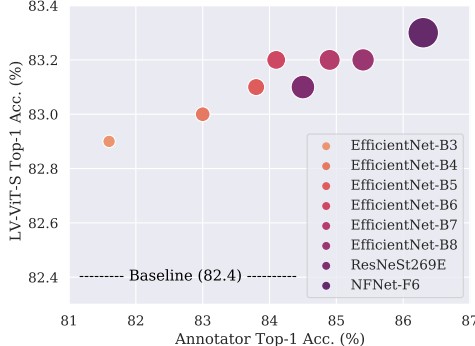

Figure 4: **Left**: LV-ViT ImageNet Top-1 Accuracy w.r.t. the token participation rate while applying token labeling. Token participation rate indicates the percentage of patch tokens involved in computing the token labeling loss. This experiment reflects that all tokens matter for vision transformers. **Right**: LV-ViT-S ImageNet Top-1 Accuracy w.r.t. different annotator models. The point size indicates the parameter number of the annotator model. Clearly, our token labeling objective is robust to different annotator models.

Table 4: Comparison of token labeling (TL), knowledge distillation (KD) based method and ReLabel method based on utilized tokens, DeiT-S/LV-ViT-S Top-1 accuracy on ImageNet validation set and training time on a single V100 GPU node.

| Method | Online KD | Online TL | TL | ReLabel | Vanilla |
|---|---|---|---|---|---|
| Tokens Utilized | 2 | All | All | 1 | 1 |
| DeiT-S Acc. (%) | 81.2 | 81.8 | 81.0 | 80.4 | 79.9 |
| LV-ViT-S Acc. (%) | 83.0 | 83.5 | 83.3 | 82.8 | 82.4 |
| Training Time (8×V100) | 63 hrs | 63 hrs | 45 hrs | 45 hrs | 41 hrs |

**Online Token Labeling:** Unlike the online knowledge distillation method which generates labels by a teacher model online, our token labeling approach utilizes the dense label map generated in advance and directly applies the corresponding augmentation methods, such as random crop, on the label map to obtain token-level labels. To directly compare with the online knowledge distillation based method and validate the effectiveness of token-level supervision, we further conduct experiments on the online version of our token labeling method, which generates token-level labels online during training. Following DeiT [36], we use RegNetY-16GF [30] as the online teacher model. Results in terms of DeiT-S/LV-ViT-S Top-1 accuracy and training time for our token labeling, online knowledge distillation, and ReLabel [49] are listed in Table 4, with number of utilized tokens also included for clear comparison. As can be seen, for both online and offline cases, using token-level supervision can improve the overall performance with only negligible additional training cost. Meanwhile, compared to the vanilla training baseline, our proposed offline token labeling brings almost no additional training cost, and boosts the overall performance of LV-ViT-S by 0.9%, which well demonstrates its efficiency and effectiveness.

**Robustness to Different Annotators:** To evaluate the robustness of our token labeling method, we use different pretrained CNNs, including EfficientNet-B3,B4,B5,B6,B7,B8 [34], NFNet-F6 [3] and ResNest269E [51], as annotator models to provide dense supervision. Results are shown in the right part of Figure 4. We can see that, even if we use an annotator with relatively lower performance, such as EfficientNet-B3 whose Top-1 accuracy is 81.6%, it can still provide multi-label location-specific supervision and help improve the performance of our LV-ViT-S model. Meanwhile, annotator models with better performance can provide more accurate supervision, bringing even better performance, as stronger annotator models can generate better token-level labels. The largest annotator NFNet-F6 [3], which has the best performance of 86.3%, allows us to achieve the best result for LV-ViT-S, which is 83.3%. In addition, we also attempt to use a better model, EfficientNet-L2 pretrained on JFT-300M as described in [49] which has 88.2% Top-1 ImageNet accuracy, as our annotator. The performance of LV-ViT-S can be further improved to 83.5%. However, to fairly compare with the models without

extra training data, we only report results based on dense supervision produced by NFNet-F6 [3] that uses only ImageNet training data.

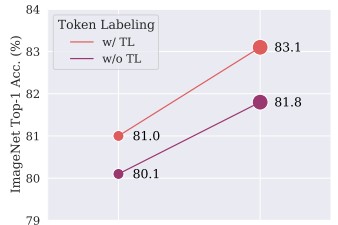 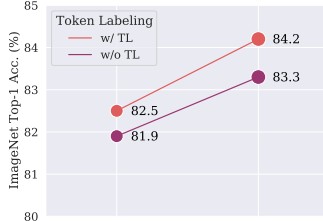 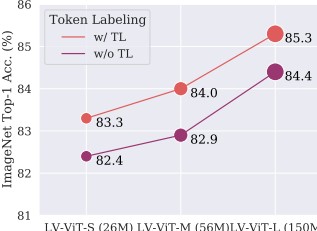

Figure 5: Performance of the proposed token labeling objective on three different vision transformers: DeiT [36] (**Left**), T2T-ViT [46] (**Middle**), and LV-ViT (**Right**). Our method has a consistent improvement on all 7 different ViT models.

**Robustness to Different ViT Variants:** To further evaluate the robustness of our token labeling, we train different transformer-based networks, including DeiT [36], T2T-ViT [3] and our model LV-ViT, with the proposed training objective. Results are shown in Figure 5. It can be found that, all the models trained with token labeling consistently outperform their vanilla counterparts, demonstrating the robustness of token labeling with respect to different variants of patch-based vision transformers. Meanwhile, for different scales of the models, the improvement is also consistent. Interestingly, we observe larger improvements for larger models. These indicate that our proposed token labeling method is widely applicable to a large range of patch-based vision transformer variants.

**Beyond Vision Transformers:** We further explore the performance of token labeling on other CNN-based and MLP-based models. Results are shown in Table 5. Besides our re-implementation with more data augmentation and regularization techniques, we also provide the results from the original papers. It can be found that for both MLP-based and CNN-based models, our token labeling objective can also improve the performance over strong baselines by providing location-specific dense supervision.

Table 5: Performance of the proposed token labeling objective on representative CNN-based (ResNeSt) and MLP-based (Mixer-MLP) models. Our method has a consistent improvement on all different models. Here $\dagger$ indicates results reported in original papers.

| Model | Mixer-S/16 [35] | | | Mixer-B/16 [35] | | | Mixer-L/16 [35] | | | ResNeSt-50 [51] | | |
|---|---|---|---|---|---|---|---|---|---|---|---|---|
| Token Labeling | ✗ | ✗ | ✓ | ✗ | ✗ | ✓ | ✗ | ✗ | ✓ | ✗ | ✗ | ✓ |
| Parameters | 18M | 18M | 18M | 59M | 59M | 59M | 207M | 207M | 207M | 27M | 27M | 27M |
| Top-1 Acc. (%) | $73.8^{\dagger}$ | 75.6 | **76.1** | $76.4^{\dagger}$ | 78.3 | **79.5** | $71.6^{\dagger}$ | 77.7 | **80.1** | $81.1^{\dagger}$ | 80.9 | **81.5** |

## 4.3 Comparison to Other Methods

We compare our proposed model LV-ViT with other state-of-the-art methods in Table 6. For small-sized models, when the test resolution is set to $224 \times 224$, we achieve an $83.3\%$ accuracy on ImageNet with only 26M parameters, which is $3.4\%$ higher than the strong baseline DeiT-S [36]. For medium-sized models, when the test resolution is set to $384 \times 384$ we achieve the performance of $85.4\%$, the same as CaiT-S36 [37], but with much less computational cost and parameters. Note that both DeiT and CaiT use knowledge distillation to improve their models, which introduce much more computations in training. However, we do not require any extra computations in training and only have to compute and store the dense score maps in advance. For large-sized models, our LV-ViT-L with a test resolution of $448 \times 448$ achieves an $86.2\%$ top-1 accuracy, which is comparable to CaiT-M36 [37] but with far fewer FLOPs and parameters.

Table 6: Top-1 accuracy comparison with other methods on ImageNet [13] and ImageNet Real [2]. All models are trained without external data. With the same computation and parameter constraint, our model consistently outperforms other CNN-based and transformer-based counterparts. The results of CNNs and ViT are referenced from [37].

| | Network | Params | FLOPs | Train size | Test size | Top-1(%) | Real Top-1 (%) |
|---|---|---|---|---|---|---|---|
| CNNs | EfficientNet-B5 [34] | 30M | 9.9B | 456 | 456 | 83.6 | 88.3 |
| | EfficientNet-B7 [34] | 66M | 37.0B | 600 | 600 | 84.3 | – |
| | Fix-EfficientNet-B8 [34, 38] | 87M | 89.5B | 672 | 800 | 85.7 | 90.0 |
| | NFNet-F3 [3] | 255M | 114.8B | 320 | 416 | 85.7 | 89.4 |
| | NFNet-F4 [3] | 316M | 215.3B | 384 | 512 | 85.9 | 89.4 |
| | NFNet-F5 [3] | 377M | 289.8B | 416 | 544 | 86.0 | 89.2 |
| Transformers | ViT-B/16 [15] | 86M | 55.4B | 224 | 384 | 77.9 | 83.6 |
| | ViT-L/16 [15] | 307M | 190.7B | 224 | 384 | 76.5 | 82.2 |
| | T2T-ViT-14 [46] | 22M | 5.2B | 224 | 224 | 81.5 | – |
| | T2T-ViT-14↑384 [46] | 22M | 17.1B | 224 | 384 | 83.3 | – |
| | CrossViT [6] | 45M | 56.6B | 224 | 480 | 84.1 | – |
| | Swin-B [25] | 88M | 47.0B | 224 | 384 | 84.2 | – |
| | TNT-B [16] | 66M | 14.1B | 224 | 224 | 82.8 | – |
| | DeepViT-S [59] | 27M | 6.2B | 224 | 224 | 82.3 | – |
| | DeepViT-L [59] | 55M | 12.5B | 224 | 224 | 83.1 | – |
| | DeiT-S [36] | 22M | 4.6B | 224 | 224 | 79.9 | 85.7 |
| | Distilled DeiT-S [36] | 22M | 4.6B | 224 | 224 | 81.2 | 86.8 |
| | DeiT-B [36] | 86M | 17.5B | 224 | 224 | 81.8 | 86.7 |
| | DeiT-B↑384 [36] | 86M | 55.4B | 224 | 384 | 83.1 | 87.7 |
| | Distilled DeiT-B [36] | 87M | 17.5B | 224 | 224 | 83.4 | 88.3 |
| | BoTNet-S1-128 [31] | 79.1M | 19.3B | 256 | 256 | 84.2 | - |
| | BoTNet-S1-128↑384 [31] | 79.1M | 45.8B | 256 | 384 | 84.7 | - |
| | CaiT-S36↑384 [37] | 68M | 48.0B | 224 | 384 | 85.4 | 89.8 |
| | CaiT-M36 [37] | 271M | 53.7B | 224 | 224 | 85.1 | 89.3 |
| | CaiT-M36↑448 [37] | 271M | 247.8B | 224 | 448 | 86.3 | 90.2 |
| Ours LV-ViT | LV-ViT-S | 26M | 6.6B | 224 | 224 | 83.3 | 88.1 |
| | LV-ViT-S↑384 | 26M | 22.2B | 224 | 384 | 84.4 | 88.9 |
| | LV-ViT-M | 56M | 16.0B | 224 | 224 | 84.1 | 88.4 |
| | LV-ViT-M↑384 | 56M | 42.2B | 224 | 384 | 85.4 | 89.5 |
| | LV-ViT-L | 150M | 59.0B | 288 | 288 | 85.3 | 89.3 |
| | LV-ViT-L↑448 | 150M | 157.2B | 288 | 448 | 85.9 | 89.7 |
| | LV-ViT-L↑448 | 150M | 157.2B | 448 | 448 | 86.2 | 89.9 |
| | LV-ViT-L↑512 | 151M | 214.8B | 448 | 512 | 86.4 | 90.1 |

## 4.4 Semantic Segmentation on ADE20K

It has been shown in [19] that different training techniques for pretrained models have different impacts on downstream tasks with dense prediction, like semantic segmentation. To demonstrate the advantage of the proposed token labeling objective on tasks with dense prediction, we apply our pretrained LV-ViT with token labeling to the semantic segmentation task.

Similar to previous work [25], we run experiments on the widely-used ADE20K [58] dataset. ADE20K contains 25K images in total, including 20K images for training, 2K images for validation and 3K images for test, and covering 150 different foreground categories. We take both FCN [26] and UperNet [44] as our segmentation frameworks and use the mmseg toolbox to implement. During training, following [25], we use the AdamW optimizer with an initial learning rate of 6e-5 and a weight decay of 0.01. We also use a linear learning schedule with a minimum learning rate of 5e-6. All models are trained on 8 GPUs and with a batch size of 16 (i.e., 2 images on each GPU). The input resolution is set to $512 \times 512$. In inference, a multi-scale test with interpolation rates of [0.75, 1.0, 1.25, 1.5, 1.75] is used. As suggested by [58], we report results in terms of both mean intersection-over-union (mIoU) and the average pixel accuracy (Pixel Acc.).

In Table 7, we test the performance of token labeling on both FCN and UperNet frameworks. The FCN framework has a light convolutional head and can directly reflect the performance of the pretrained models in terms of transferable capability. As can be seen, pretrained models with token

Table 7: Transfer performance of the proposed LV-ViT in semantic segmentation. We take two classic methods, FCN and UperNet, as segmentation architectures and show both single-scale (SS) and multi-scale (MS) results on the validation set.

| Method | Token Labeling | Model Size | mIoU (SS) | P. Acc. (SS) | mIoU (MS) | P. Acc. (MS) |
|---|---|---|---|---|---|---|
| LV-ViT-S + FCN | ✗ | 30M | 46.1 | 81.9 | 47.3 | 82.6 |
| LV-ViT-S + FCN | ✓ | 30M | 47.2 | 82.4 | 48.4 | 83.0 |
| LV-ViT-S + UperNet | ✗ | 44M | 46.5 | 82.1 | 47.6 | 82.7 |
| LV-ViT-S + UperNet | ✓ | 44M | 47.9 | 82.6 | 48.6 | 83.1 |

labeling perform better than those without token labeling. This indicates token labeling is indeed beneficial to semantic segmentation.

We also compare our segmentation results with previous state-of-the-art segmentation methods in Table 8. Without pretraining on large-scale datasets such as ImageNet-22K, our LV-ViT-M with the UperNet segmentation architecture achieves an mIoU score of 50.6 with only 77M parameters. This result is much better than the previous CNN-based and transformer-based models. Furthermore, using our LV-ViT-L as the pretrained model yields a better result of 51.8 in terms of mIoU. As far as we know, this is the best result reported on ADE20K with no pretraining on ImageNet-22K or other large-scale datasets.

Table 8: Comparison with previous work on ADE20K validation set. As far as we know, our LV-ViT-L + UperNet achieves the best result on ADE20K with only ImageNet-1K as training data in pretraining. [†]Pretrained on ImageNet-22K.

| | Backbone | Segmentation Architecture | Model Size | mIoU (MS) | Pixel Acc. (MS) |
|---|---|---|---|---|---|
| CNNs | ResNet-269 | PSPNet [54] | - | 44.9 | 81.7 |
| | ResNet-101 | UperNet [44] | 86M | 44.9 | - |
| | ResNet-101 | Strip Pooling [22] | - | 45.6 | 82.1 |
| | ResNeSt200 | DeepLabV3+ [8] | 88M | 48.4 | - |
| Transformers | DeiT-S | UperNet | 52M | 44.0 | - |
| | ViT-Large[†] | SETR [56] | 308M | 50.3 | 83.5 |
| | Swin-T [25] | UperNet | 60M | 46.1 | - |
| | Swin-S [25] | UperNet | 81M | 49.3 | - |
| | Swin-B [25] | UperNet | 121M | 49.7 | - |
| | Swin-B[†] [25] | UperNet | 121M | 51.6 | - |
| LV-ViT | LV-ViT-S | FCN | 30M | 48.4 | 83.0 |
| | LV-ViT-S | UperNet | 44M | 48.6 | 83.1 |
| | LV-ViT-M | UperNet | 77M | 50.6 | 83.5 |
| | LV-ViT-L | UperNet | 209M | **51.8** | **84.1** |

## 5 Conclusions and Discussion

In this paper, we introduce a new token labeling method to help improve the performance of vision transformers. We also analyze the effectiveness and robustness of our token labeling with respect to different annotators and different variants of patch-based vision transformers. By applying token labeling, our proposed LV-ViT achieves 84.4% Top-1 accuracy with only 26M parameters and 86.4% Top-1 accuracy with 150M parameters on ImageNet-1K benchmark.

Despite the effectiveness, token labeling has a limitation of requiring a pretrained model as the machine annotator. Fortunately, the machine annotating procedure can be done in advance to avoid introducing extra computational cost in training. This makes our method quite different from knowledge distillation methods that rely on online teaching. For users with limited machine resources on hand, our token labeling provides a promising training technique to improve the performance of vision transformers.

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
