# 1  Experiment Details

We show the default hyper-parameters for our ImageNet classification experiments in Table 1. In addition, for fine-tuning on larger image resolution, we set batch size to 512, learning rate to 5e-6, weight decay to 1e-8 and fine-tune 30 epochs. Other hyper-parameters are set the same as default. During training, a machine node with 8 NVIDIA V100 GPUs (32G memory) is required. When fine-tuning our large model with image resolution of $448 \times 448$, we need 4 machine nodes with the same GPU settings as above.

Table 1: Default hyper-parameters for our experiments. Note that we do not use the MixUp augmentation method when ReLabel or token labeling is used.

| Supervision | Standard | ReLabel | Token labeling |
|---|---|---|---|
| Epoch | 300 | 300 | 300 |
| Batch size | 1024 | 1024 | 1024 |
| LR | $1\text{e-}3 \cdot \frac{\text{batch\_size}}{1024}$ | $1\text{e-}3 \cdot \frac{\text{batch\_size}}{1024}$ | $1\text{e-}3 \cdot \frac{\text{batch\_size}}{640}$ |
| LR decay | cosine | cosine | cosine |
| Weight decay | 0.05 | 0.05 | 0.05 |
| Warmup epochs | 5 | 5 | 5 |
| Dropout | 0 | 0 | 0 |
| Stoch. Depth | 0.1 | 0.1 | 0.1 |
| MixUp alpha | 0.8 | - | - |
| Erasing prob. | 0.25 | 0.25 | 0.25 |
| RandAug | 9/0.5 | 9/0.5 | 9/0.5 |

# 2  More Experiments

## 2.1  Training Technique Analysis

We present a summary of our modification and proposed token labeling method to improve vision transformer models in Table 2. We take the DeiT-Small [4] model as our baseline and show the performance increment as more training techniqeus are added. In this subsection, we will ablate the proposed modifications and evaluate the effectiveness of them.

Table 2: Ablation path from the DeiT-Small [4] baseline to our LV-ViT-S. All experiments expect for larger input resolution can be finished within 3 days using a single server node with 8 V100 GPUs. Clearly, with only 26M learnable parameters, the performance can be boosted from 79.9 to 84.4 (**+4.5**) using the proposed Token Labeling and other proposed training techniques.

| Training techniques | #Param. | Top-1 Acc. (%) |
|---|---|---|
| Baseline (DeiT-Small [4]) | 22M | 79.9 |
| + More transformers ($12 \rightarrow 16$) | 28M | 81.2 (**+1.2**) |
| + Less MLP expansion ratio ($4 \rightarrow 3$) | 25M | 81.1 (**+1.1**) |
| + More convs for patch embedding | 26M | 82.2 (**+2.3**) |
| + Enhanced residual connection | 26M | 82.4 (**+2.5**) |
| + Token labeling with MixToken | 26M | 83.3 (**+3.4**) |
| + Input resolution ($224 \rightarrow 384$) | 26M | 84.4 (**+4.5**) |

**Explicit inductive bias for patch embedding:** Ablation analysis of patch embedding is presented in Table 3. The baseline is set to the same as the setting as presented in the third row of Table 2. Clearly, by adding more convolutional layers and narrow the kernel size in the patch embedding, we can see a consistent increase in the performance comparing to the original single-layer patch embedding. However, when further increasing the number of convolutional layer in patch embedding to 6, we do not observe any performance gain. This indicates that using 4-layer convolutions in patch embedding is enough. Meanwhile, if we use a larger stride to reduce the size of the feature map, we can largely reduce the computation cost, but the performance also drops. Thus, we only apply a

convolution of stride 2 and kernel size 7 at the beginning of the patch embedding module, followed by two convolutional layers with stride 1 and kernel size 3. The feature map is finally tokenized to a sequence of tokens using a convolutional layer of stride 8 and kernel size 8 (see the fifth line in Table 3).

Table 3: Ablation on patch embedding. Baseline is set as 16 layer ViT with embedding size 384 and MLP expansion ratio of 3. All convolutional layers except the last block have 64 filters. #Convs indicatie the total number of convolutions for patch embedding, while the kernel size and stride correspond to each layer are shown as a list in the table.

| #Convs | Kerenl size | Stride | Params | Top-1 Acc. (%) |
|---|---|---|---|---|
| 1 | [16] | [16] | 25M | 81.1 |
| 2 | [7,8] | [2,8] | 25M | 81.4 |
| 3 | [7,3,8] | [2,2,4] | 25M | 81.4 |
| 3 | [7,3,8] | [2,1,8] | 26M | 81.9 |
| 4 | [7,3,3,8] | [2,1,1,8] | 26M | **82.2** |
| 6 | [7,3,3,3,3,8] | [2,1,1,1,1,8] | 26M | **82.2** |

**Enhanced residual connection:** We found that introducing a residual scaling factor can also bring benefit as shown in Table 4. We found that using smaller scaling factor can lead to better performance and faster convergence. Part of the reason is that more information can be preserved in the main branch, leading to less information loss and better performance.

Table 4: Ablation on enhancing residual connection by applying a scaling factor. Baseline is a 16-layer vision transformer with 4-layer convolutional patch embedding. Here, function $F$ represents either self-attention (SA) or feed forward (FF).

| Forward Function | #Parameters | Top-1 Acc. (%) |
|---|---|---|
| $X \longleftarrow X + F(X)$ | 26M | 82.2 |
| $X \longleftarrow X + F(X)/2$ | 26M | **82.4** |
| $X \longleftarrow X + F(X)/3$ | 26M | **82.4** |

**Larger input resolution:** To adapt our model to larger input image, we interpolate the positional encoding and fine-tune the model on larger image resolution for a few epochs. Token labeling objective as well as MixToken are also used during fine-tuning. As can be seen from Table 2, fine-tuning on larger input resolution of $384 \times 384$ can improve the performance by $1.1\%$ for our LV-ViT-S model.

## 2.2 Comparison with CaiT

CaiT [5] is currently the best transformer-based model. We list the comparison of training hyperparameters and model configuration with CaiT in Table 5. It can be seen that using less training techniques, computations, and smaller model size, our LV-ViT achieves identical result to the state-of-the-art CaiT model.

Table 5: Comparison with CaiT [5]. Our model exploits less training techniques, model size, and computations but achieve identical result to CaiT.

| Settings | LV-ViT (Ours) | CaiT [5] |
|---|---|---|
| Transformer Blocks | 20 | 36 |
| #Head in Self-attention | 8 | 12 |
| MLP Expansion Ratio | 3 | 4 |
| Embedding Dimension | 512 | 384 |
| Stochastic Depth [3] | 0.2 (Linear) | 0.2 (Fixed) |
| Rand Augmentation [2] | ✓ | ✓ |
| CutMix Augmentation [6] | | ✓ |
| MixUp Augmentation [7] | | ✓ |
| LayerScaling [5] | | ✓ |
| Class Attention [5] | | ✓ |
| Knowledge Distillation | | ✓ |
| Enhanced Residuals (Ours) | ✓ | |
| MixToken (Ours) | ✓ | |
| Token Labeling (Ours) | ✓ | |
| Test Resolution | $384 \times 384$ | $384 \times 384$ |
| Model Size | 56M | 69M |
| Computations | 42B | 48B |
| Training Epoch | 300 | 400 |
| ImageNet Top-1 Acc. | 85.4 | 85.4 |

# 3  Visualization

We apply the method proposed in [1] to visualize both DeiT-base and our LV-ViT-S. Results are shown in Figure 1 and Figure 2. In Figure 1, we can observe that our LV-ViT-S model performs better in locating the target objects and hence yields better classification performance with high confidence. In Figure 2, we visualize the top-2 classes predicted by the two models. Noted that we follow [1] to select images with at least 2 classes existing. It can be seen that our LV-ViT-S trained with token labeling can accurately locate both classes while the DeiT-base sometimes fails in locating the entire target object for a certain class. This demonstrates that our token labeling objective does help in improving models' visual grounding capability because of the location-specific token-level information.

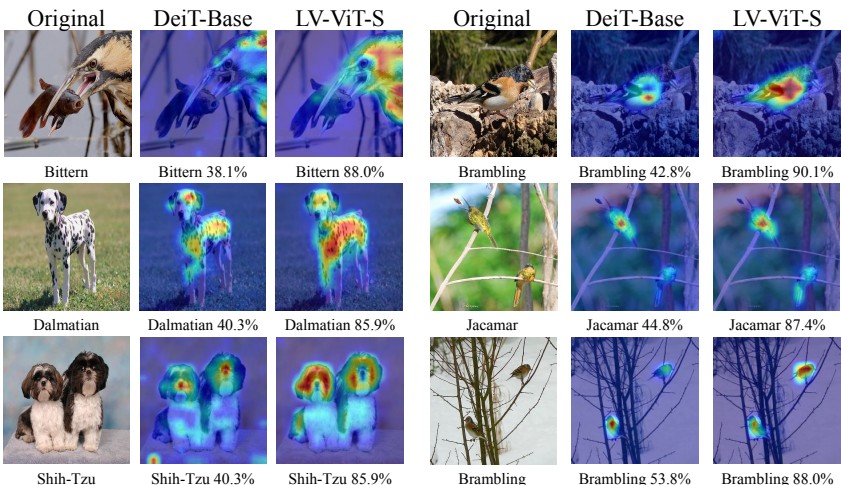

Figure 1: Visual comparisons between DeiT-base and LV-ViT-S.

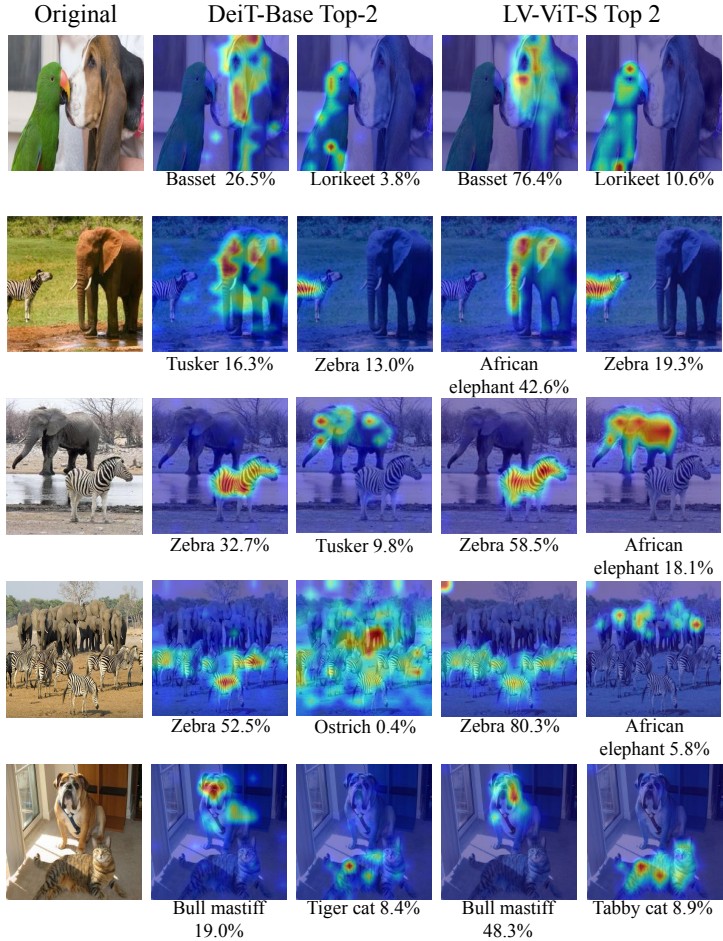

Figure 2: Visual comparisons between DeiT-base and LV-ViT-S for the top-2 predicted classes.