# OpenReview forum: "All Tokens Matter: Token Labeling for Training Better Vision Transformers"
_NeurIPS.cc/2021/Conference — NeurIPS 2021 Poster_

### Official Review · Reviewer_mqsb · 2021-07-08

**Rating:** 7
**Confidence:** 4

**Summary:**

This paper introduces token labelling, a new training objective for vision transformers. The core idea is to supervise each of the patches of the last layer, instead of simply discarding them as is usually done. The proposed approach significantly improves performance of various architectures on ImageNet.

**Limitations And Societal Impact:**

Yes

**Main Review:**

Overall, I like the idea presented in this paper, which is simple and yields good results. Various ablation convincingly demonstrate the effectiveness and wide applicability of the token labelling technique. However, I am not sure I understood the claims made by the authors that this approach is more efficient computationally than distillation, as it requires annotating each one of the patches. However, I still think this paper deserves publication.

Strengths:
- The proposed approach reaches state-of-the art, both in classification and segmentation
- Extensive ablations effectively demonstrate the benefit of token labelling
- Although this is not mentioned in the main text (likely due to last minute rush), a section in the appendix shows that token labelling also benefits MLP-mixer architectures. If the paper is accepted, I believe this section should go in the main text, as it is important to show the wide applicability of this method (perhaps replacing the ablation over the annotator which is less crucial)

Weaknesses:
- Storing the score maps is costly. This is acknowledged by the authors, but they seem to suggest nonetheless that token labelling is computationally cheaper than distillation. I do not understand this, see questions below.
- It would be nice to include latency or throughput measurements to give an idea of the speed of the models considered, and check that fetching the score maps from memory doesn’t slow down training.

Comments:
- “Unlike knowledge distillation methods that require a teacher model to generate supervision labels online, token labeling is a cheap operation. The dense score map can be generated by a pretrained model in advance “. I do not understand this sentence : token labelling involves a forward pass of an annotator in the same way as distillation does, what is the difference ? The distillation tokens can also be stored in advance, or am I missing something ?
- I am confused as to why the score maps are 18x18x1000 rather than just 18x18. Does this mean the score maps are soft labels ? Couldn’t it be possible to store simply 18x18 maps of correct labels, and supervise with hard labels ? Since Touvron et al. find that hard label distillation is better than soft label distillation for DeiT, perhaps the same could hold true for token labelling ?
- In the comparison table, I think it would be fair to add a comparison with the distilled DeiT and CaiT models, since token labelling is rather similar in spirit to distillation. Ideally, it would even be nice to have, among the ablations, a direct comparison of token labelling vs token distillation in the same training setting.

**Time Spent Reviewing:**

4

---

> ### Author Response · Authors · 2021-08-10
> **Response to Reviewer mqsb**
>
> Thanks for your valuable and detailed comments.
>
> > Q1. The difference between knowledge distillation and token labeling.
>
> A1. We summarize the difference between online knowledge distillation (online KD), online token labeling (online TL), relabel and token labeling(TL) as follows.
>
> |          Method              | online KD | online TL | Relabel |   TL  | Vanilla |
> | :---------------------------- | :---------: | :---------: | :---------: | :--------: | :------: |
> | Augment-aware label |      Y      |      Y      |      Y      |     Y      |     N     |
> | LV-ViT ImageNet Acc.|     83.0   |    83.5   |   82.8    |   83.3   |   82.4   |
> |Training time (8xV100)|   63 hrs  |   63 hrs  |  45 hrs  |  45 hrs | 41 hrs |
> | # Utilized  Tokens       |       2      |     All     |      1       |    All     |      1    |
>
> **Augmentation-aware label:**  Different from the vanilla training pipeline where each image has a fixed label, online distillation based method and our method will assign different labels for the same image according to the different data augmentation such as random crop. While the online knowledge distillation method involves a forward pass during training, our Token labeling method directly applies the same spatial augmentation to the dense score map to obtain the token level label for the augmented image. This is much more efficient, and brings negligible computation cost.
>
> **Performance:**  Compared to the Relabel method, our token labeling further improves the utilization rate of token features by assigning a dense location-specific token-level label. While the relabel method can improve the performance of LV-ViT-S by 0.4 point, the dense token-level label can help improve the LV-ViT model by 0.9 point.
>
> **Training efficiency:** Compared to the online knowledge distillation method which generates augmentation aware labels by a teacher model online, our token labeling method utilizes the dense label map generated in advance and directly applies the corresponding spatial augmentation such as random crop on the label map to obtain token-level labels. Thus, our token labeling method is more efficient than the online knowledge distillation based method (63 hrs vs 45 hrs on a single node with 8 V100 GPUs) and is more friendly for those with limited computation resources.
>
> **Number of tokens utilized for supervision:** While methods like online knowledge distillation and Relabel use one or two tokens for loss computation, our proposed token labeling method uses all the patch tokens as well as an additional class token for training. Experimental results also show that using all tokens for supervision can consistently improve the model performance.
>
> **Summary:** Both the result of offline token labeling and online token labeling suggest that dense supervision is helpful for training vision transformer models. To the best of our knowledge, we are the first to utilize all tokens for training image classification models and prove its effectiveness on both classification accuracy as well as performance on downstream tasks like segmentation.
>
> > Q2. It would be nice to include latency or throughput measurements to give an idea of the speed of the models considered, and check that fetching the score maps from memory doesn’t slow down training.
>
> A2. We list the training throughput on a single node (8xV100)  with respect to different supervision such as token labeling, online knowledge distillation and online token labeling. As shown in the table below, the online knowledge distillation based method is much slower and has much higher training cost, while our token labeling is much more efficient ( 1.5x faster than online KD for DeiT-S).
>
> |   Model   |    online KD    |    online TL    | Token labeling | Vanilla training |
> | :----------- | :---------------:  | :----------------: |-------------------: | :------------------: |
> | LV-ViT-S |   1698.2 im/s  |   1703.3 im/s |     2375.9 im/s |     2612.2 im/s   |
> |   DeiT-S   |   2088.9 im/s  |   2081.3 im/s |    3271.6 im/s  |     3696.8 im/s  |
>
>
> > Q3. Why the score maps are 18x18x1000 rather than just 18x18. Does this mean the score maps are soft labels?
>
> A3. Yes, we use soft labels as supervision and thus the score maps are 18x18x1000. Following the reviewer’s suggestion, we provide results of using soft-label and hard-label in the table below. As suggested by our experiment and the relabel paper [47], using soft labels and keeping top 5 entries for each token-level label gives the best result. So we only store a 18x18x5 value-index map for top 5 entries at each location and map it to a 18x18x1000 soft (sparse) label map (which will further be cropped and resized to 14x14x1000) while generating labels for training. Due to the difference between the label generating procedure of online distillation and offline methods like Relabel and our Token labeling, using hard labels for our method performs worse. We will make this clear in the revised version.
>
> |               Top K                |    1    |    2    |    3    |   4    |   5    |  10  |
> | :------------------------------- | :------ | :-----: | :-----: |:-----: | :----: | ----: |
> | LV-ViT-S ImageNet Acc.|  83.0  | 83.0 |  83.2 | 83.3 | 83.3 | 83.2 |
>
> (ImageNet Top-1 Accuracy with respect to top k entries kept in the label map.)
>
> > Q4. In the comparison table, it would be fair to add a comparison with the distilled DeiT and CaiT models.
>
> A4. We thank the reviewer for this useful suggestion. We have included CaiT models trained using hard knowledge distillation as supervision in Table 4, and we will highlight this in the revised version. Due to the limited space, a complete comparison between CaiT-S36 and our model has been listed in Section 2.3 in the appendix. As can be seen, with much fewer FLOPs and parameters, our LV-ViT still performs comparably with state-of-the-art models like CaiT trained using online knowledge distillation. We will add the results of models with hard knowledge distillation listed below to Table 4 as references.
>
> | Model              | Param |FLOPs | Top1 Acc.|
> | :------------------- | -------: |-------: | ------------: |
> | DeiT-S              |  22M   | 4.6B  |     79.9     |
> | Distilled DeiT-S|  22M   | 4.6B  |      81.2     |
> | DeiT-B              |  86M   |17.5B |      81.8     |
> | Distilled DeiT-B |  87M  | 17.5B |     83.4     |
> | LV-ViT-S            |  27M  | 6.6B  |      83.3     |
>
>
> > Q5. A direct comparison of token labelling vs token distillation in the same training setting.
>
> A5. We gratefully thank the reviewer for this valuable comment. We would like to highlight that after submission, we indeed did experiments on online token labeling. The following table shows the relevant results.
>
> | Model     | online TL | online KD | Token labeling | Vanilla training |
> | :----------- | :----------: | :-----------: | :------------------: | :------------------: |
> | DeiT-S    |    81.8     |     81.2      |         81.0         |         80.1          |
> | LV-ViT-S |    83.5     |      83.0     |         83.3         |         82.4          |
>
> As can be seen from the above, online token labeling improves the performance by 0.5-0.6 point upon the knowledge distillation baseline using the same RegNetY-16GF as teacher model, showing the advantage of token level supervision. We will add the above comparison of online knowledge distillation and online token labeling into the ablation study in the revised version.

---

> > ### Comment · Reviewer_mqsb · 2021-08-11
> > **Thank you for the clarification**
> >
> > Thanks for your response !
> > With your explanation of augmentation-aware labels, I now understand why token labelling can be done in advance rather than online.
> > This was less clear in the main text, and could perhaps be emphasized more.
> > I think the table comparing the throughputs also helps to show the efficiency of this method compared to online methods.

---

### Official Review · Reviewer_kkHf · 2021-07-10

**Rating:** 6
**Confidence:** 5

**Summary:**

This paper presents a novel approach for training vision transformer. Instead of using the image-level label as the supervision, the proposed approach uses pixel-level labels as supervision, and use the cutmix-like image augmentation (MixToken) to generate pixel-level labels. The proposed approach can be regarded as the combination of cutmix [46] and relabeling [47]. Experimental results implied that proposed scheme improves the training quality.

**Limitations And Societal Impact:**

The main limitation of this paper is the limited novelty. The proposed MixToken and Token Labeling can be viewed as a modification of cutmix [46] and relabel [47]. The modification leads to $0.5$ gain (Table 2), which is not significant.

**Main Review:**

Originality: The main drawback of this paper is that the novelty is limited. The proposed MixToken and Token Labeling can be viewed as a modification of cutmix [46] and relabel [47]. The modification leads to $0.5$ gain (Table 2), which is not significant.

Quality and clarity: this paper is clearly written, and the reader can easily understand this paper. The experiments are also very strong.

Significance:  Experimental results show that the proposed schemes are helpful. Unfortunately, this paper is more like a straightforward application of cutmix and relabel with slight modifications. This reduces the technical signification.

**Time Spent Reviewing:**

3.5 hours

---

> ### Author Response · Authors · 2021-08-10
> **Response to Reviewer kkHf**
>
> Thanks a lot for reviewing our paper and we address your concerns as follows.
>
> > Q1. The proposed MixToken and Token Labeling can be viewed as a modification of cutmix [46] and relabel [47].
>
> A1.  Token labeling and MixToken are quite different from the original Relabel [47] and CutMix [46] in the following aspects. The Relabel method and knowledge distillation-based methods rely on image-level labels as global supervision. Differently, our Token Labeling reformulates the original training of the image classification as token level dense prediction and uses token-level labels as supervision to better train the vision transformer models. As explained in Section 3.2 of our paper, this training strategy provides location-specific information from the score maps generated by the annotators, which can aid the training models to improve recognition accuracy. In addition, our major contribution lies in demonstrating that all tokens contribute to the classification performance of vision transformers. For MixToken, we use it as a substitution for CutMix while applying token labeling in order to provide a more accurate token-level label. Combining the proposed methods together, we can further improve the performance over strong cutmix+relabel baseline with negligible computation cost as shown in Table 2 of the main paper.
>
> Moreover, We also conduct experiments to evaluate our token labeling method in the online setting and compare it with online knowledge distillation (KD). The following table shows the relevant results.
>
> |   Model   | online KD | online TL | Vanilla training |
> | :----------- | :----------- | :------------: | :-----------------: |
> |   DeiT-S   |    81.2      |    81.8       |         80.1        |
> | LV-ViT-S |    83.0      |    83.5       |         82.4         |
>
> As can be seen from the above, online token labeling improves the performance by 0.5-0.6 point upon the knowledge distillation baseline, showing the advantage of token level supervision.
>
> > Q2. The modification leads to 0.5  gain (Table 2), which is not significant.
>
> A2. As shown in the following table, our baseline result by LV-ViT-S has already attained 82.4 top-1 accuracy, which we believe is a rather high number compared to previous CNN-based or transformer-based models with the similar model size constraint. Given this baseline, our Token Labeling can further improve the result to 83.3 (+0.9), 0.5 better than that with Relabel, which we believe is already a large improvement. After submission, we also attempted to see the performance improvement brought by token labeling over Relabel on more popular ViT models (e.g., T2T-ViT and DeiT) as shown below.
>
> |   Model   | Token labeling | Relabel | Vanilla training |
> | :----------  | :-----------------: |:---------: | :------------------: |
> | LV-ViT-S |         83.3         |   82.8    |         82.4         |
> |T2T-ViT-7|         73.3         |   72.3    |         71.7         |
> |   DeiT-S   |         81.0         |   80.4    |         79.9         |
> |   DeiT-B   |         83.1         |   82.5    |         81.8         |
>
>
> From the above results, we can conclude that Token labeling consistently outperforms the relabel method by a notable margin. Note that the original Relabel method [47] relies on a strong model trained with extra data, while in our case, we limit the training data to ImageNet 1k and use NFNet-F6 as the annotator to train vision transformers. In this setting, the relabel method only has marginal improvements upon the baseline ( e.g. 0.4 point for LV-ViT-S and 0.5 for T2T-ViT). However, by applying our proposed token labeling method, the performance gain is boosted to 0.9/1.6 over the vanilla training baseline for LV-ViT-S/ T2T-ViT-7, which shows that token level supervision is much more effective.

---

> > ### Comment · Reviewer_kkHf · 2021-08-12
> > **Acceptable, but incremental**
> >
> > Thanks a lot for providing feedbacks to me.
> >
> > Actually, the two feedbacks still did not convince me. MixToken is a slight modification of CutMix: only patch only comes from a single image other than possibly two images. Token Labeling is a dense prediction manner, and might be not done for supervised Pretraining, but already studied in related work, such as contrastive learning. In addition, it is not sure if the MixToken and Token Labeling can be used for CNNs and if the performance gain is similar. IMO, similar gain for CNNs can be achieved. If similar gain is achieved, the word "token" in the two terms, MixToken, and Token Labeling might simplified as Patch or something similar.
> >
> > Considering great performance achieved by the authors' hard work, I upgraded the rating.
> >
> > I also would like to hear the feedbacks about my comments from other reviewers and the ACs.

---

### Official Review · Reviewer_tac1 · 2021-07-16

**Rating:** 7
**Confidence:** 5

**Summary:**

This paper presents a new strategy to more effectively train vision transformers by introducing token-wise supervision. A token labeling loss, MixToken data augmentation and several modifications on ViT architectures are proposed. Extensive experiments are conducted to show the effectiveness of the proposed method.

**Limitations And Societal Impact:**

The limitations and potential negative societal impact of this paper have been discussed.

**Main Review:**

The proposed token labeling method is well motivated and extensively validated. The proposed method can largely improve the performance of vision transformers while maintaining the relatively simple architecture (only the patch embedding layer is replaced by a small CNN). The modifications on the original ViT may also be useful in future research and applications.  Extensive experiments are conducted to show the effectiveness of the proposed method. The new designs on the network archietcture are clearly verified. It is interesting to see that the token labeling method also works well on DeiT, T2T-ViT, Mixer and CNNs. However, I still have some concerns:

- The proposed token labeling method is very similar to hard knowledge distillation and the method is closely related to ReLabel strategy [47]. Using the knowledge distillation method to vision transformers is also studied in DeiT [34]. Although I agree with the authors that the proposed method is more efficient, the technical novelty of the proposed method is limited. Besides, I think it would be better to provide the results of online hard/soft knowledge distillation as references.

- I think it is necessary to provide a more detailed analysis of the differences between the distillation method proposed in DeiT and the proposed token labeling. According to Figure 5 (left), the top-1 accuracy of the DeiT-B model with token labeling is 83.1%, while [34] reports that DeiT-B with hard distillation (300 epochs) can achieve 83.4% top-1 accuracy. It seems that the simple hard distillation method in DeiT can outperform token labeling. However, the performance gap may come from different teacher models or the online/offline distillation strategy. To support the motivation of token labeling, it is critical to show the proposed token-wise supervision is more effective than global supervision like DeiT-B. I think the authors can provide the results of online token labeling or use NFNet-F6 as the teacher model for global distillation to directly compare with the distillation method in DeiT. I suspect token-wise supervision may be more useful for downstream dense prediction tasks. This paper can be much stronger if the advantages of token labeling (compared to global distillation in DeiT) are clearly demonstrated.

- Although I agree the results are impressive, I still think emphasizing the performance under a certain level of #Param may not be very reasonable. With similar parameters, models using larger input images will always lead to better #Param/accuracy trade-offs. I think FLOP is a better metric to measure the computational complexity and the throughput/latency are even more realistic metrics. It would be better to report the FLOPs/throughput/latency for the main results and comparisons.

Overall, I think this is a practical method with several technical innovations and good results. However, the proposed token labeling method may not be clearly verified. As its current state, I would like to rate this paper as Marginally below the acceptance threshold. I will be happy to upgrade my rating if my second concern is properly addressed.

**Time Spent Reviewing:**

5 hours

---

> ### Author Response · Authors · 2021-08-10
> **Response to Reviewer tac1**
>
> Thanks for your valuable and detailed comments.
>
> > Q1.It would be better to provide the results of online hard/soft knowledge distillation as references.
>
> A1. We thank the reviewer for this useful suggestion. We have included CaiT models trained using hard knowledge distillation as supervision in Table 4, and we will highlight this in the revised version. Due to the limited space, a complete comparison between CaiT-S36 and our model has been listed in Section 2.3 in the appendix. As can be seen, with much fewer FLOPs and parameters, our LV-ViT still performs comparably with state-of-the-art models like CaiT trained using online knowledge distillation. We will add the results of models with hard knowledge distillation listed below to Table 4 as references.
>
> | Model              | Param |FLOPs | Top1 Acc.|
> | :------------------- | :-------: |-------: | ------------: |
> | DeiT-S              |  22M   | 4.6B  |     79.9     |
> | Distilled DeiT-S|  22M   | 4.6B  |      81.2     |
> | DeiT-B              |  86M   |17.5B |      81.8     |
> | Distilled DeiT-B |  87M  | 17.5B |     83.4     |
> | LV-ViT-S            |  27M  | 6.6B  |      83.3     |
>
> > Q2. Provide the results of online token labeling.
>
> A2. We gratefully thank the reviewer for this valuable comment. We would like to highlight that after submission, we indeed did experiments on online token labeling. The following table shows the relevant results.
>
>
> | Model     | online TL | online KD | Token labeling | Vanilla training |
> | :----------- | :----------: | :-----------: | :------------------: | :------------------: |
> | DeiT-S    |    81.8     |     81.2      |         81.0         |         80.1          |
> | LV-ViT-S |    83.5     |      83.0     |         83.3         |         82.4          |
>
> As can be seen from the above, online token labeling improves the performance by 0.5-0.6 point upon the knowledge distillation baseline using the same RegNetY-16GF as teacher model, showing the advantage of token level supervision. We will add the above comparison of online knowledge distillation and online token labeling into the ablation study in the revised version.
>
> > Q3. It would be better to report the FLOPs/throughput/latency for the main results and comparisons.
>
> A3. We have reported parameters and FLOPs in Table 4 for comparison of computation cost. We have tested the inference-time throughput as follows and we will add these results as references in the revised version. We can see that with fewer FLOPs and faster inference speed, our LV-ViT-M outperforms the distilled DeiT-B model, indicating the effectiveness of our token labeling method and proposed model.
>
> (Table. Model inference throughput tested on a single V100 GPU with batch size 32.)
>
> |   Model   | Param | FLOPs | image size | throughput | Top-1 Acc. |
> | :----------- | ---------: | --------: | :------------: |--------------: | :-------------: |
> | LV-ViT-S |   26M    |  6.6B    |       224    | 1018.2 im/s |    83.3       |
> | LV-ViT-M |  56M    |  16.0B   |       224    |  668.9 im/s |     84.1      |
> | LV-ViT-L |   150M  |   59.0B  |       288    |  204.8 im/s |    85.3      |
> | Distilled DeiT-S | 22M  | 4.6B   |   224    |  1321.1 im/s |    81.2     |
> | Distilled DeiT-B | 87M  | 17.5B |   224    |  659.1 im/s   |    83.4     |

---

> > ### Comment · Reviewer_tac1 · 2021-08-12
> > **Thanks for your respones**
> >
> > Thank you for the detailed feedback and the new results. I am pleased to see online token labeling can outperform online hard distillation. I think the new results can clearly support the motivation of token labeling. The comparisons based on throughput also help to demonstrate the efficiency of the proposed method. Overall, I believe this is a solid work with several technical innovations and good results. Therefore, I would like to upgrade my rating to Accept.

---

### Decision · Program_Chairs · 2021-09-27

**Decision:**

Accept (Poster)

**Comment:**

This paper proposed a new framework to train vision transformers with token level supervision. While there were some debate over the novelty of the proposed technique in the context of previous methods (MixToken, CutMix, relabel), it was agreed that the proposed method is efficient and performant, and the extensive empirical studies in this paper could benefit the community. The authors also did well by providing convincing results on extra baselines requested by the reviewers in the rebuttal.